# Human Aldehyde Dehydrogenases: A Superfamily of Similar Yet Different Proteins Highly Related to Cancer

**DOI:** 10.3390/cancers15174419

**Published:** 2023-09-04

**Authors:** Vasileios Xanthis, Theodora Mantso, Anna Dimtsi, Aglaia Pappa, Vasiliki E. Fadouloglou

**Affiliations:** Department of Molecular Biology & Genetics, Democritus University of Thrace, 68100 Alexandroupolis, Greece

**Keywords:** aldehyde dehydrogenases (ALDHs), cancer, chemotherapy resistance, crystallins, inhibitors, 3D structure, post-translational modifications, topology, quaternary association

## Abstract

**Simple Summary:**

Cancer is found amongst the leading causes of death globally, with its incidence rates expected to increase even more over the next decades. Human aldehyde dehydrogenases (hALDHs) are members of the superfamily of NAD(P) dependent enzymes responsible for the oxidation of a variety of endogenous and exogenous aldehydes to their corresponding carboxylic acids. Interestingly, several members of the superfamily have been implicated in cancer pathology. This review provides a detailed description of their multiple physiological functions and 3D structures, and explains their roles in cancer pathology and chemotherapy resistance. It also discusses the effect of structural features, variations and/or alterations on the enzymes’ function, and capacity to interact with other proteins. Overall, we aim to provide a better understanding of ALDHs role in cancer development and the promising effects of their inhibition in cancer therapy.

**Abstract:**

The superfamily of human aldehyde dehydrogenases (hALDHs) consists of 19 isoenzymes which are critical for several physiological and biosynthetic processes and play a major role in the organism’s detoxification via the NAD(P) dependent oxidation of numerous endogenous and exogenous aldehyde substrates to their corresponding carboxylic acids. Over the last decades, ALDHs have been the subject of several studies as it was revealed that their differential expression patterns in various cancer types are associated either with carcinogenesis or promotion of cell survival. Here, we attempt to provide a thorough review of hALDHs’ diverse functions and 3D structures with particular emphasis on their role in cancer pathology and resistance to chemotherapy. We are especially interested in findings regarding the association of structural features and their changes with effects on enzymes’ functionalities. Moreover, we provide an updated outline of the hALDHs inhibitors utilized in experimental or clinical settings for cancer therapy. Overall, this review aims to provide a better understanding of the impact of ALDHs in cancer pathology and therapy from a structural perspective.

## 1. Introduction

Aldehydes are ubiquitous in the environment and massively involved in biological processes. In a living cell, aldehydes can have either endogenous or exogenous origin. Naturally produced aldehydes are intermediates and byproducts of metabolic pathways of substances such as hydrocarbons, amines, amino acids, alcohols, vitamins, steroids, etc. They can also be produced as responses to environmental stresses such as cellular lipid peroxidation (LPO). On the other hand, exogenous aldehydes can be introduced to the organism as constituents of plant- and animal-based foods or site products of cooking processes. Nowadays, human activities have additionally increased the exogenous aldehyde load: junk food, highly processed foods, refined oils, cigarette smoke, and environmental pollution include high concentrations of aldehydes [1,2,3].

Aldehydes and their derivatives are usually deleterious for the cell because they are highly reactive chemical reagents. They can attack and damage DNA and proteins by degrading these biomolecules, forming adducts or inducing mutations and conformational changes. Apart from the eliminated or impaired functions of the affected macromolecules, an increased oxidative stress induction through radical oxygen species (ROS) production and LPO is also observed [4,5,6,7]. Accumulated aldehydes in the cell have been correlated with apoptosis, impaired cellular homeostasis and mitochondrial respiration, and carcinogenesis [8,9]. As a defense cellular mechanism, biochemical pathways have evolved to metabolize aldehydes and ensure normal living conditions. One of these pathways involves aldehyde oxidation by the enzymatic activity of aldehyde dehydrogenases (ALDHs). ALDHs, which are NAD(P)-dependent enzymes, have been identified in many organisms such as bacteria, fungi, plants, and mammals and are responsible for aldehyde conversion to their corresponding carboxylic acids, which are much better tolerated by the cells [10,11,12].

ALDHs show a broad substrate specificity being able to process saturated and unsaturated, straight and branched-chain aliphatic and aromatic aldehydes of various lengths. In humans, there are 19 ALDH isoforms (hALDHs) which are distributed in different tissues and cell types and display different substrate specificities. In particular, their known substrates include aldehydes involved in growth and development, differentiation, oxidative stress, osmoregulation, and neurotransmission as well as dietary and environmental aldehydes. Furthermore, ALDHs are important for cell survival because of their antioxidant potential to absorb UV light, scavenge free radicals through their methionine and cysteine sulfhydryl groups, and bind a variety of molecules such as cholesterol and androgen [1,13,14,15]. Consequently, ALDHs are significant in human biology because they are involved in defense and detoxification mechanisms and numerous biosynthetic, metabolic, and signal transduction pathways related to pathogenicity and health conditions such as neurogenerative diseases, cancer, and resistance to chemotherapy. For instance, ALDH1 isoforms participate in retinoic acid cell signaling, by oxidizing irreversibly all-trans- and 9-cis retinal to all-trans- and 9-cis retinoic acid (RA) in a tissue-specific reaction [16,17,18]. RA is essential for development, apoptosis, and differentiation by binding to retinoic acid receptor α (RARα), retinoic X receptor (RXR), nuclear hormone receptors, and peroxisome proliferator activated receptor beta/delta (PPAR/β/δ), which are transcription factors that induce the transcriptional activity of 500 target genes [18,19]. In cancer, RA binds to RARα and alternative transcriptional partners are recruited into the nucleus and enhance cell proliferation, drug resistance, and inhibition of apoptosis by activating c-MYC, cyclin D1, and ALDH1A1 [20,21].

Today, all the 19 hALDHs have known amino acid sequences, while 12 of them have determined 3D structures and the substrate specificities are well characterized for several of them. In addition, their relation to carcinogenesis and chemotherapy resistance as well as their significance as biomarkers and drug targets are well documented [22]. Here, we attempt a comprehensive review of the superfamily concerning (i) their multiple physiological functions, (ii) their relation to disease, especially focusing on their roles in cancer pathology and resistance to chemotherapies as well as (iii) current advances in therapeutic treatments based on hALDH inhibition. In some cases, their pathological functions can be directly related to structural changes occurring either on proteins’ quaternary assembly level or on the substrate and cofactor binding sites as consequences of amino acid mutations, post-translational modifications, etc. We discuss how such changes may affect the proteins’ catalytic efficiency and their binding affinity for interactions with molecular partners. Moreover, we show that division in families and subfamilies based on amino acid sequences is also strongly founded on structural differences. Although the superfamily adopts a generally similar folding pattern [11], differences in topology as well as in the tertiary and quaternary level of structural organization may explain their great functional variability.

## 2. The hALDH Protein Superfamily: Structure Description, Comparison, and Classification

ALDH enzymes are present in all three taxonomic domains (Archaea, Eubacteria, and Eukarya), therefore supporting their fundamental significance throughout evolutionary history. Because they fulfil significant roles in many processes of life, they have attracted scientific interest for several years. For instance, some of the first high-resolution 3D structures of ALDH enzymes were determined in the 1990s and belong to class3 *rattus norvegicus* ALDH [23], the class2 beef mitochondria ALDH [24], and the class1 sheep liver ALDH [25].

The human Aldehyde Dehydrogenase (hALDH) superfamily consists of 19 genes which encode 19 enzyme isoforms. Based on their sequences, the hALDHs are traditionally grouped into families (≥40% sequence identity) and subfamilies (≥60% sequence identity). The proteins have been accordingly named as hALDH1-9, 16, and 18 [26,27]. The hALDH2 protein was originally given a different number-name, although we now know that it is quite similar with members of the ALDH1 family and together they constitute the ALDH1/2 family (Table 1).

Today, 12 hALDHs have determined 3D structures and 105 released PDB codes are related to them (until 1 June 2023). A comparison of these structures shows that although the different members may be assembled in different quaternary associations to form biologically relevant particles (see Table 1), the monomers are organized in a quite similar way (Figure 1). Each chain consists of three distinct domains, i.e., a NAD(P)-binding domain, a catalytic domain, and an oligomerization domain (Figure 1A–C). The NAD(P)-binding domain and the catalytic domain are both Rossmann folds (Figure 1D,E). Each of the substrate and cofactor binding site pockets are funnel-shaped passages which have in common their narrow bottoms forming a double funnel (Figure 1B). There, in the narrowest area of the double funnel is the interface of substrate and cofactor binding passages where the catalytic residues are located (Figure 1). All hALDHs except hALDH16A1 pseudoenzyme (see also below) incorporate a group of highly conserved residues which are important for enzymatic activity (Figure 2). A cysteine residue plays the role of a catalytic thiol. A glutamic acid activates a water molecule which exerts thiol deprotonation and activation of cysteine nucleophile. The activated cysteine can then attack the substrate’s aldehyde group and forms an oxyanion thiohemiacetal intermediate, stabilized in part by an asparagine. The negatively charged oxygen of the oxyanion intermediate then facilitates hydride transfer to the NAD(P) cofactor, resulting in the formation of a thioacylenzyme intermediate. Hydrolysis of the thioacylenzyme and release of carboxylic acid product takes place via the same glutamic acid. The mechanism assumes that the reduced cofactor dissociates afterwards. The only known exception so far is ALDH6A1 (Figure 2) which follows a slightly different mechanism, and the reduced cofactor is released prior to the deacylation step. Consequently, the reaction’s product is a CoA ester instead of a free carboxylic acid [1,28,29].

Although hALDHs share significant sequence and structural conservations, they do have differences which are determinant factors of their variability in enzymatic specificities and binding affinities for a great number of substrates and small or larger molecules, as we present with details in the following sections. Figure 3 shows a structural comparison and classification of the hALDH monomers performed using tools of DALI server [30]. It is evident that the structural classification agrees well with the sequence classification, even though it reveals additional features and structure-based subgroups. Thus, the structural analysis separates hALDHs into three major groups (Figure 3). The structure of ALDH18A1 deviates significantly from all the rest; hALDH3A1 and hALDH3A2 form a second independent group and the remaining proteins belong to a third group. Within the third group, ALDH1/2 family members form a compact subgroup; ALDH2 and ALDH1B1 are the closest relatives while ALDH1A2 is the most distant member of the subgroup (Figure 3). Topological analysis of the monomers reveals some of the structural determinants for the grouping mentioned above. Figure 4 displays the topological diagrams for each of the three domains of the 12 hALDHs of known structure. All but hALDH18A1 have in common at least the topology of the catalytic domain. All three domains of hALDH18A1 have distinct topologies in comparison with the other members of the superfamily. This is the reason why hALDH18A1 deviates so significantly from the superfamily. hALDH3A1 and 3A2 have in common the topology of cofactor binding domain, which is quite distinct from the other members (Figure 4). hALDH1/2, 5A1, and 9A1 have common topologies in all three domains, while hALDH4A1 and 7A1 might be the bridge between the 3A subgroup and the 1/2, 5A1, 9A1 subgroup.

The above highlighted differences in the primary structure and topology level become profound on the tertiary level since they impose packing geometry differences on the monomers and a distinct distribution of domains on the dimers’ surface (Figure 5). There are also consequences on the quaternary association level of individual members. The different topologies of oligomerization domains underline the potential of various quaternary assemblies within the superfamily members. Indeed, comparison of oligomerization domain topology classification presented in Figure 4 and data presented in Table 1 clearly indicate a correlation between different topologies and observed oligomerization states. Thus, members of the 1/2, 5, 7, and 9 families usually adopt a tetrameric state, while 3Ai and 4A1 proteins are dimers. In addition, hALDH18A1 with significantly different topology seems also to be a dimer. Furthermore, a possible correlation between redox and oligomerization state has also been observed for the hALDH5A1 enzyme (Figure 6).

## 3. The Multiple Physiological Roles of hALDHs

ALDH isoforms are widely distributed throughout the human body, indicative of their critical roles in multiple processes. Liver, kidneys, heart, and brain as well as tissues which are rich in mitochondria have the highest mRNA and protein expression of ALDHs. Similarly, hALDHs have a wide subcellular distribution related to their specific function. They are found in the cytoplasm, mitochondrion, endoplasmic reticulum (ER), and nucleus [1,32,33]. Details of their organ/tissue distribution and subcellular localization are included in Table 1.

Interestingly, several members of the ALDH superfamily are moonlighting proteins, meaning they exhibit multiple cellular roles and diverse biochemical and biophysical functions [34]. In particular, catalytic ALDHs with a second role as crystallins were among the first proteins to be characterized as dual-function, moonlighting proteins. Crystallins are present in abundance in the lens and cornea of the eye, and some of them are identical to ALDH classes 1, 2, and 3 [35]. For example, the omega crystalline found in scallop, squid, and octopus and the eta crystalline found in elephant shrews is the same protein as aldehyde dehydrogenase enzyme [36,37,38,39]. The first human enzyme isoforms to be recognized as corneal crystallins were ALDH3A1 and ALDH1A1 [33,40]. 

The multiple functions of hALDHs are summarized in Figure 7 and analytically presented below.

### 3.1. hALDHs Are Active Enzymes

hALDHs are enzymes with at least three catalytic activities and multiple substrates. The different enzymatic activities of the family are summarized below.

#### 3.1.1. Aldehyde Dehydrogenases

hALDHs have as primary enzymatic activity that of dehydrogenase on a wide range of aldehyde substrates. This function requires NAD(P) as cofactor. During catalysis, the aldehyde is oxidized to the corresponding carboxylic acid and the cofactor is reduced to NAD(P)H. As is explained in detail below, the aldehyde dehydrogenase catalytic activity makes hALDHs (i) major cellular detoxification factors and (ii) key components in the biosynthetic pathways of significant compounds.

Aldehyde dehydrogenase activity is directly involved in aldehyde metabolism and halts their buildup in cells. The cellular aldehyde content originates from either exogenous or endogenous aldehydes [26]. Endogenous aldehydes can stem from normal metabolism of assorted biomolecules, namely, amino acids, biogenic amines, vitamins, lipids, or steroids, while the most common sources of exogenous aldehydes are metabolized drugs, nutritional components, and environmental substances such as fume, smog, or cigarette smoke [41]. In their majority, aldehydes, either endogenous or exogenous, are highly reactive and therefore highly toxic species with cytotoxic and potentially carcinogenic effects. Therefore, the aldehyde dehydrogenase activity establishes hALDHs as major cellular detoxification factors utilized by the human organism to counteract the accumulation of aldehydes and their harmful effects [5].

Although ALDHs are considered mainly as detoxifying enzymes, their role expands in a variety of often unexpected biological processes. A particularly important function of hALDHs is their participation in the synthesis of important biomolecules that originate from the metabolism of aldehydes and/or their intermediates [27]. These biomolecules include betaine (see below ‘osmotic pressure regulators’ paragraph), folate, γ-aminobutyric acid (GABA), and most prominently, RA. RA is the product of the irreversible oxidation of retinal, which is mediated by members of the hALDH1A family (hALDH1Ai) (see Table 1) [27,42]. This indicates the importance of the ALDH proteins in RA signaling and therefore the expression of genes necessary for growth and development [42]. The significance of certain ALDH isoenzymes in RA signaling has been confirmed by several scientific reports. More specifically, it has been shown that the knockout of ALDH1A2 and ALDH1A3 in mice resulted in death during the embryonic and early development stages due to abnormal organ development [43,44]. Furthermore, the rescue of ALDH mice was possible through treatment with RA, indicating that the main role of ALDHs in embryogenesis and development is the biosynthesis of RA and not the direct promotion of gene expression [43].

Another important function of ALDHs is their role in NAD(P)H synthesis as the result of NAD(P) cofactor reduction. Reduced nicotinamide adenine dinucleotide phosphate NAD(P)H is a critical electron donor in all organisms and plays at least two important roles. Firstly, by supplying electrons it contributes to multiple biological reactions as the biosynthesis of fatty acids, cholesterol, steroids, and deoxynucleotides, and of macromolecules such as nucleic acids and lipids. Secondly, NAD(P)H is involved in maintaining cellular homeostasis by affecting the redox balance within the cell [28]. In particular, NAD(P)H participates in the regeneration of reduced glutathione (GSH) from its oxidized form (GSSG) via the glutathione reductase/peroxidase system [45]. Glutathione exists in cells in a ratio of oxidized (GSSG) and reduced (GSH) forms and when the ratio of GSSG-to-GSH is elevated, there is redox imbalance and therefore induction of oxidative stress that can eventually cause cellular damage. Moreover, NAD(P)H may act as a direct antioxidant of radicals [46,47].

Additionally, NAD(P)H provides electrons to O_2_ leading to the generation of H_2_O_2_ or superoxide by the action of NADPH oxidases [48,49,50,51]. Furthermore, NAD(P)H supports the function of several P450 enzymes as it acts as a reducing power, therefore allowing them to complete the detoxification of various metabolites, xenobiotics, etc.

#### 3.1.2. Esterases

Apart from their well-known dehydrogenase activity, several hALDHs—such as ALDH1A1, ALDH2, ALDH3A1, and ALDH4A1—have a secondary role as esterases, i.e., catalyze ester hydrolysis (p-nitrophenyl esters) [1,15]. Early studies focusing on investigating the catalytic properties of ALDHs from different organisms in liver tissues have monitored the hydrolysis of p-nitrophenyl acetate (p-NPA) as well as long-chain carboxylic acid p-nitrophenyl esters [52]. Evidence supporting the esterase activity of ALDHs was studied even earlier, in the 1970s, by using samples from horse liver tissue for kinetic experiments [53]. It is worth mentioning that ALDHs utilize the same active site pocket and the same catalytic residues for both dehydrogenase and esterase functions: the highly reactive and conserved cysteines i.e., Cys302 for ALDH1A1 and ALDH2, Cys243 for ALDH3A1, etc. (Figure 2). Nevertheless, a cofactor, NAD(P) or other, is not required for the esterase activity [28,29].

#### 3.1.3. Nitrate Reductases

A third catalytic function, that of nitrate reductase, has been described in the literature for mammalian ALDH2. In particular, it has been shown that ALDH2 can process nitroglycerin (GTN) and produces 1,2-glyceryl dinitrite and nitrite, which in turn get converted to NO which stimulates soluble guanylated cyclase (sGC) followed by an increase in cGMP. When ALDH2 inhibitors were used, the hypotensive effects of GTN were diminished in vivo, whereas addition of the nitrosovasodilator sodium nitroprusside led to vasorelaxation. Consequently, it was proposed that ALDH2 reductase capacity caused the formation of cGMP and relaxation of vascular smooth muscle in vitro (murine ALDH2) as well as in vivo (rabbit ALDH2) [54]. Later, a novel and very sensitive genetically encoded fluorescent probe for NO was used in order to monitor its generation intracellularly in response to low GTN exposure. The involvement of hALDH2 in the GTN bioactivation was confirmed by using mutant (with reduced denitration activity but retaining NO generation capacity) and wild-type hALDH2 from vascular smooth muscle cells (VSMCs). It was found that there was significant formation of NO at a greater extent in the mutant compared to wild-type hALDH2 VSMCs which was accompanied by the activation of purified soluble guanylated cyclase (sGC) in cell lysates and elevated cGMP levels following administration of GTN. These observations indicated that hALDH2 takes part in the bioactivation of GTN in mutant cells and is related to NO generation. In fact, the increased formation of NO and reduced levels of 1,2-GDN demonstrated a shift from the clearance-based GTN denitration to the NO pathway, therefore explaining how low doses of GTN can still be effective through the hALDH2-catalysis of NO pathway. To further support these findings, when ALDH2 inhibitors were utilized, NO generation and increase in cGMP levels were prevented, thus demonstrating the implication of ALDH2 in the formation of NO, which is responsible for the observed relaxation in VSMCs, through the bioactivation of GTN [55]. Structural evidence from a triple-mutant of low denitration capacity of hALDH2 in complex with GTN confirmed that GTN is bound to the dehydrogenase catalytic site of the enzyme. Moreover, it showed that the process of GTN denitration is triggered by the nucleophilic attack of the catalytic Cys302 at a terminal nitrate group, while a thionitrate intermediate is formed followed by the already known formation of 1,2-glyceryl dinitrate. In addition, MS data supported a disulfide or a sulfinic acid in the catalytic site of hALDH2 under reversible and irreversible inhibition of its denitration activity [56].

Other ALDHs that possess the nitrate reductase function have been suggested to be ALDH1A1 and ALDH1B1 in mice, rabbits, and humans [27,33,57]

### 3.2. ALDHs Are Pseudoenzymes

An intriguing member of the hALDH superfamily is ALDH16A1 which appears to be an inactive enzyme. Sequence alignment (Figure 2) and crystallographic data have revealed that hALDH16A1 lacks the whole catalytic triad; therefore, it is incapable of performing any of the dehydrogenase or esterase activities and it is characterized as a pseudoenzyme [58]. Pseudoenzymes are proteins incapable of exerting enzymatic activity, despite their significant sequence and structural similarities with active enzymes. The lack of a few but catalytically important residues is usually the reason why these proteins are enzymatically inactive, as is the case of ALDH16A1. There is growing evidence that pseudoenzymes maintain and evolve significant functions other than enzymatic catalysis. For instance, they act as allosteric regulators, binding scaffolds and regulators of conventional enzymes by competing for the same substrate or by participating in the assembly of the holoenzyme [59,60,61].

Regarding ALDH16A1’s function, there is evidence that it interacts with maspardin protein, which is responsible for the pathogenesis of the mast syndrome (SPG21) when it is truncated. However, it remains unclear what effects this interaction has on maspardin and whether it is associated with the pathogenic phenotype [62]. According to a recent report using *Aldh16a1* knockout mice, it was shown that ALDH16A1 has a functional role in the kidney as it caused differential expression of numerous genes and affected cellular lipid and lipid metabolism processes as well [63]. Two spliced variants have been identified in humans (ALDH16A1, long form and ALDH16A1_v2, short form). Results from a study utilizing whole-genome sequencing in a group of Icelanders revealed a rare missense single nucleotide polymorphism (SNP) in *Aldh16a1* and it was associated with gout and hyperuricemia. More specifically, a cytosine is substituted by a guanine in exon 13 (c.1580C>G) of the 17-exon transcript and in exon 12 (c.1427C>G) of the 16-exon transcript of *Aldh16a1*, therefore causing a missense proline to arginine change of amino acids 527 (p.Pro527Arg) and 476 (p.Pro476Arg), respectively [64]. The hypothesis that ALDH16A1 interacts with hypoxanthine-guanine phosphoribosyltransferase (HPRT1), which is implicated in the metabolism of uric acid and gout, was supported by the fact that the carriers of the variant present hyperuricemia. Therefore, it was suggested that the interaction between wild-type ALDH16A1 and HPRT1 has a beneficial effect on HPRT1 function, whereas when this interaction is obstructed, it potentially contributes to the overproduction of uric acid which is observed in the variant carriers [33].

In conclusion, there are a few scientific reports demonstrating that although ALDH16A1 is enzymatically inactive, it retains significant biological functions, which are based on its ability to interact with multiple proteins. Nevertheless, the exact biological roles of this isoform are not fully understood yet and remain to be characterized regarding its involvement in pathological conditions.

### 3.3. ALDHs Are Molecular Chaperones

#### 3.3.1. Crystallins and Anti-Stress Protein Factors

Another interesting and unique role of ALDHs is represented by the function of ALDH3A1 and ALDH1A1 isoforms as corneal and lens crystallins, respectively [33]. In mammalian corneal cells, ALDH3A1 can account for up to half of the cell’s water-soluble protein and has been found to have a fundamental role in the transparency and the refractory properties of the cornea along with ALDH1A1 [65,66].

Apart from their structural function as crystallins, ALDH1A1 and ALDH3A1 also exhibit antioxidant capacity and have a protective role against oxidative stress induced by ultra-violet (UV) light and other factors [67]. This is achieved by various mechanisms such as direct absorption of UV radiation through a suicide response, scavenging the produced ROS and other free radicals, metabolizing toxic aldehydes that are produced during lipid peroxidation by ROS, preventing protein misfolding through a chaperone-like function, and generating NAD(P)H indirectly [45,68,69,70]. UV radiation can be harmful to ocular tissue cells mainly because of the formation of free radicals, resulting in pathological eye conditions such as cataracts and retinal or corneal degeneration [71]. However, when human corneal epithelial cells transfected with ALDH3A1 were compared to mock-transfected cells (which did not express the protein) they were found to be better protected against oxidative-stress induced by UV radiation and 4-HNE [72].

The protective role of ALDH3A1 on other proteins, which are exposed to various stress conditions such as thermal and chemical stresses, has also been assessed. In particular, ALDH3A1 appeared to protect restriction enzyme SmaI and citrate synthase under thermal stress. Furthermore, human corneal epithelial cells stably transfected with the enzyme were more resistant to cytotoxicity against chemical stressors such as H_2_O_2_ and tert-butyl hydroperoxide [73].

ALDH2 has also been associated with oxidative stress-related pathological conditions in other tissues. For example, impairment of ALDH2 activity has been related to oxidative stress induction in experimental animal models focusing on their cardiovascular system [32,74].

#### 3.3.2. Osmotic Pressure Regulators

An interesting additional function of hALDHs involves the regulation of osmotic pressure. hALDH7A1 appears to resemble the green garden pea ’26 g protein’ by sharing 60% sequence identity. The 26 g protein is a modulator of cellular osmotic pressure and prevents oxidative stress in response to draught conditions when its expression levels increase. Similarly, ALDH7A1 protects the cell against hyperosmotic stress by the formation of osmolyte betaine from betaine aldehyde and metabolizing other LPO-derived aldehydes [42,75,76].

### 3.4. ALDHs Are Binding Scaffolds

Findings from various studies on vertebrate ALDHs have highlighted the capacity of certain isoforms to act as binding proteins for endogenous and exogenous substances as well. For instance, mouse ALDH2 (mALDH2) was found to bind acetaminophen, a commonly used antipyretic and analgesic [26,77], while hALDH1A1 can bind androgen [78], xenopus, hALDH1A1 thyroid hormone [79,80], and cholesterol [26]. Moreover, mouse and human ALDH1A1 isoforms present binding capacity for exogenously produced quinolone [81], mALDH1A1 for daunorubicin [82], and hALDH1A1 for flavopiridol [26,33,83,84]. As far as androgen is concerned, results from a research study demonstrated that the expression of mALDHA1 in Leydig cells is regulated in a developmental pattern in the testis of mice. Leydig cells produce androgens which are needed for the differentiation of the reproductive tract in males. It was suggested that mALDH1A1 expression could be mediated by androgen receptors in Leydig cells; then, mALDH1A1 could act by oxidizing retinal to RA which, in turn, would regulate testosterone synthesis [85]. However, the physiological role of the observed interactions remains to be elucidated [1]. There is no structural evidence showing how the above-mentioned substances bind the enzyme and whether this binding interferes with the enzyme’s catalytic function. However, it has been proposed that the cofactor binding domain may be involved in these interactions [84].

## 4. hALDHs Are Associated with Chronic Diseases and Conditions

ALDHs have a strong relation to a variety of serious human diseases and conditions. The majority of those are genetic metabolic disorders and a variety of cancer types [18,86].

Τhe Sjögren-Larsson syndrome (SLS) is a neurological condition that follows an autosomal recessive pattern of inheritance and its genetic basis is a wide variety of mutations in the *Aldh3a2* gene, which encodes the fatty aldehyde dehydrogenase (FALDH, Table 1) [87]. The mutated enzyme loses up to 90% of its activity resulting in impairing metabolism of fatty alcohols and their accumulation in cells. This seems to disrupt the normal membrane formation as well as the myelin formation which, in turn, accounts for the clinical symptoms of SLS, namely, cutaneous abnormality (ichthyosis or scaly skin), gradually increasing spasticity, delayed development, etc. [87,88].

The alcohol flushing syndrome especially common in Asian populations, which is described as a sensitivity to consuming alcohol accompanied by headache, nausea, and a distinctive face flushing, has its genetic base in an inherited mitochondrial ALDH2 deficiency [89]. The deficiency is caused by a mutated allele of the *Aldh2* gene (ALDH2*2), identified as the Glu487Lys point mutation leading in reduced enzyme activity [90]. Consequently, a dysfunction in ethanol digestion is observed because the catabolic pathway is hindered at the stage of acetaldehyde metabolism to the less cytotoxic acetate, which is the reaction catalyzed by the ALDH2 enzyme. The aggregation of the cytotoxic, mutagenic, and carcinogenic acetaldehyde in human cells seems to be the cause of most clinical symptoms that accompany ALDH2 deficiency [91]. Heterozygotes are less affected by the mutated allele, but they still exhibit a pathological phenotype due to the participation of both mutated and wild-type peptide chains in the ALDH2 tetramer assembly [92]. Over the last decades, it was shown that this ALDH2 deficiency is correlated with a higher risk of esophageal cancer and other cancer types of the upper digestive tract (due to the increased concentration of the carcinogenic acetaldehyde) [93,94].

## 5. hALDHs Are Key Players in Cancer Pathology

ALDHs’ abnormal expression seems to have a dual role in cancer pathology by either leading to carcinogenesis or protecting cancer cell populations, especially cancer stem cells (CSC), against chemotherapy [95]. Insufficient ALDH activity results in carcinogenesis due to aldehyde accumulation. Aldehydes are highly reactive species (see above for a detailed description of their impact) and attack biomolecules of significant importance, such as nucleic acids, proteins, membranes, lipids, etc., causing their inactivation and leading to disruption of crucial cellular functions [96,97,98]. 

On the other hand, high ALDH activity in CSC has been correlated with their survival and antioxidant profile [41,99]. hALDHs are highly expressed in normal stem cell populations such as hemopoietic, mammary, intestinal, neural, and prostate cell populations [100,101,102,103,104,105], enhancing their abilities of self-protection, expansion, and differentiation [106]. Likewise, ALDH overexpression in CSC populations offers significant survival advantages and even chemotherapy resistance; therefore, their overexpression is correlated with poor clinical outcome and ALDHs are often used as cancer markers. 

A summary of current knowledge and most recent research, which associates abnormal expression of specific hALDH isoforms with particular cancer types, is given below.

***Blood cancer.*** It is well established that hematopoietic stem cells (HSCs) as well as their cancerous counterparts’ leukemia stem cells (LSCs) overexpress hALDH proteins. Consequently, the clarification of the family’s significance and the role of individual members in leukemia initiation and development is of great interest [107,108]. 

hALDH1A1 and hALDH3A1 affect ROS and reactive aldehyde’s metabolism in HSCs and they have been related to leukemia initiation [6,109]. Furthermore, a reduction in the quantity of HSCs and bone marrow stem cells was observed upon the knockout of ALDH1A1 and ALDH3A1 in adult mouse bone marrow cells [107,109,110]. Moreover, transduction of NUP98-HOXA10 homeodomain fusion protein in murine model leads to HSC expansion without malignant transformation, while the same transduction in *Aldh1a1/3a1−/−* murine HSCs promotes the development of leukemia [111,112]. Experiments in murine HSCs showed that ALDH1A1 is not the predominantly expressed isoform, thus laying the foundation that many other ALDHs play a role in HSC biology. Ιn particular, *Aldh9a1* exhibits the highest expression level, followed by *Aldh2*, *Aldh1a1,* and then *Aldh3a2* and *Aldh1a7* [113]. Interestingly, when *Aldh3a2* is depleted, leukemia cells die through an iron-dependent oxidative process, while normal hematopoiesis is unaffected [114].

***Breast cancer.*** Breast cancer patient prognosis is poor when ALDH1A1’s activity is high [101]. It has been shown that the Notch signaling pathway plays a special role in inducing carcinogenesis and tumor growth by re-activating post-translationally inactivated ALDH1A1. In particular, Notch pathway induces expression of sirtuin 2 (SIRT2) which deacetylates modified Lys353 on ALDH1A1’s surface (Figure 8), therefore converting the protein to its active form [115,116,117,118]. Furthermore, it has been shown that NANOG signaling increases ALDH1A3 activity by activating the NOTCH1 and AKT pathways, which stimulates DNA double-strand break repair capability and confers radio-resistance to breast cancer cell lines [119]. In a recent study, samples from patients with breast cancer were analyzed by immunohistochemistry (IHC) and the results correlated high expression of ALDH2 to poor prognosis. In particular, ALDH2 elevated expression in combination with low expression of IGSF9 and PRDM16 was linked to advanced clinicopathological features, and shorter overall survival and disease-free survival [120]. Altogether, these findings demonstrate the significance of ALDH1A1, ALDH1A3, and ALDH2 inhibition as potential therapeutic targets in breast cancer.

***Oral cancer***. ALDH1A3 and ALDH3A1 were the main ALDH isoforms in healthy oral mucosa keratinocytes collected from tissue adjacent to the wisdom teeth, as was shown by ALDEFLUOR assay, IHC analyses, and in situ hybridization of mRNA [121]. ALDH1A3 was also expressed by all three cancer cell lines tested by Hedberg et al. (2001), i.e., cultured primary keratinocytes, immortalized oral cell line SVpgC2a, and oral squamous cancer cell line SqCC/Y1 [122]. ALDH1A1 and ALDH3A1 were only present in the oral squamous cancer cell line, whereas ALDH3A2, ALDH4A1, ALDH7A1, and ALDH9A1 had varied expression patterns [122]. A recent work which focused on studying ALDH7A1 expression patterns gave controversial results when comparing data from in vitro and in vivo assays [123]. In this study, in vitro silencing of ALDH7A1 decreased cell migration, whereas overexpressing ALDH7A1 increased cell migration. However, comparison of ALDH7A1 expression in normal and tumor tissues showed that it was lower in tumor tissues than in normal ones.

***Colorectal cancer.*** Although early evidence had supported that ALDH1A1 is the dominant isoform in colorectal cancer and may also be employed as a marker for colorectal CSCs [124,125], recent research confirms that different colorectal cancer cell lines might express different hALDHs as predominant isoforms. ALDEFLUOR+ cells from colorectal cancer tissues seemed to exhibit stem cell traits and were capable of forming primary tumors in immunodeficient mice, in contrast to ALDEFLUOR− cells [102]. Over one third of colorectal tumor cells had high ALDH1A1 expression and were localized in crypts resembling the spatial distribution of healthy stem cells [125]. In athymic mice, the tumorigenicity of HT-29 cells was reduced by ALDH1A1 silencing [126]. The ALDH expression pattern of colon cancer-derived cell lines is as follows; LS-180 cells express ALDH1A1 and ALDH2 isoforms, HT-29 cells mostly express ALDH1A1 isoforms, and HCT-116 cells primarily express ALDH1A3 isoforms [126]. 

Slim columnar cells in undifferentiated normal colon tissue were shown to exhibit a minor quantity of ALDH1B1 and ALDH1A1 proteins. While no ALDH expression was visible in differentiated or stromal cells, ALDH1B1 was highly expressed in cells close to the sides of the crypt bottom and sporadically expressed in cells on both sides of the upper portion of the crypt. This cellular pattern closely resembles the spatial distribution of stem cells. ALDH1B1 was expressed at high levels in 97.5% of the colonic adenocarcinoma samples, whereas ALDH1A1 was detected at relatively low levels in only 36.6% of the samples. As a result, ALDH1B1 may serve as yet another marker for colorectal cancer [127]. Along with this, there is evidence linking ALDH1B1 with colorectal cancer stemness, resulting in chemotherapy resistance [128].

***Pancreatic cancer***. In pancreatic cancer, ALDEFLUOR+ cells are about 3% of the cells and have CSC traits [129,130]. Comparison of the ALDH1A1 mRNA expression pattern in seven pancreatic cancer cell lines (BxPC3, T3M4, PANC1, SU8686, Colo-357, AsPC-1, and MiaPaCa-2) with one non-cancerous pancreatic cell line (ACBRI) showed that BxPC3, T3M4, and PANC1 exhibited minimal to no ALDH1A1 mRNA expression, whereas SU8686 and Colo-357 cells showed moderately low levels, and AsPC-1 and MiaPaCa-2 cells exhibited nine- and three-fold ALDH1A1 mRNA overexpression compared with the ACBRI cells [131]. Moreover, gene expression analysis of the pancreatic cancer cell lines AsPC-1, BxPC3, and MiaPaCa-2 showed that AsPC-1 and BxPC3 cells mainly express ALDH1A3, while MiaPaCa-2 cells express ALDH1A1 and ALDH1A3 in equal levels [132]. 

Another study combining data derived from The Cancer Genome Atlas Program (TCGA) and The Genotype-Tissue Expression (GTEx) project, showed that 1A3, 1B1, 2, 3A1, 3B1, 4A1, 7A1, and 9A1 hALDH isoforms exhibit increased expression in pancreatic cancer. However, only the high expression of 3A1, 3B1, and 7A1 were linked to a poor outcome for individuals with pancreatic cancer. Of the three, ALDH7A1 is the most prevalent isoform in pancreatic ductal adenocarcinoma, as Western blot analysis revealed. Moreover, it was shown that gossypol and phenformin, which inhibit ALDH7A1 and oxidative phosphorylation, can prevent tumor growth in the KPC mouse model and the xenograft mouse model [133]. 

ALDH1L2 was also highly expressed in patients with pancreatic ductal adenocarcinoma, along with methylenetetrahydrofolate dehydrogenase 2 (MTHFD2) and serine hydroxymethyltransferase (SHMT2), and was evaluated as a potent marker for overall survival and disease-free survival [134].

***Lung cancer.*** Non-small cell lung cancer (NSCLC) cells with ALDH activity showed high clonogenicity, invasiveness, and chemoresistance. In addition, ALDH overexpression in specimens derived from patients with stage I NSCLC was correlated with poor prognosis [135]. ALDH1A1 and ALDH3A1 knockdown can reduce clonogenicity and motility and increase sensitivity to 4-hydroperoxycyclophosphamide in NSCLC [136,137]. Given that ALDH1A1 expression is associated with Notch transcription, pharmacological or genetic interference with Notch can decrease ALDH1A1’s activity in lung CSCs [138,139]. Furthermore, NFATc2/SOX2 coupling can upregulate ALDH1A1 expression, reduce oxidative stress from cancer drug treatment, and increase resistance to chemotherapy and targeted therapy [140]. SOX9 also seems to endorse stemness and induce chemoresistance in NSCLC cells by activating ALDH1A1 expression [141]. ALDH1A1 knockdown inhibited the invasive ability and tumorigenicity of ALDEFLUOR+ cells, indicating that it is the major isoform in lung cancer [142]. Nevertheless, other studies suggest that ALDH1A3 might be the main isoform because of its relative high expression in NSCLC cell lines. ALDH1A3 knockdown reduced ALDEFLUOR+ cells, colony formation, and tumorigenicity [143]. In A549 cells, inhibition of ALDH3A1 and ALDH1A1, by using lentiviral-mediated expression of specific siRNA, resulted in a 95% decrease of ALDH activity and reduced colony formation and migration [144].

Other ALDH isoforms found in lung cancer and proposed to be markers of poor prognosis or cancer recurrence are ALDH1L2 [145], ALDH3B1 [146,147], and ALDH7A1 [148,149]. Additionally, ALDH18A1 seems to be overexpressed in lung cancer tissues compared with normal lung tissues [150] and is upregulated in NSCLC cell lines, A549 and the lung fibroblast cell lines, SV-80 ALDH18A1, PLOD2, and P4HA1 [151]. However, recent studies using antibodies, for both Western blotting and IHC, are not conclusive on whether ALDH18A1 is expressed in NSLC cells and KRASLA2 mouse tissue [152,153,154]. Recently, it was shown that ALDH1L1 is highly expressed on mRNA level in the gefitinib-resistant human lung adenocarcinoma HCC-827/GR cells [155]. 

***Melanoma.*** ALDH1A subfamily expression in melanoma regulates CSC proliferation, apoptosis, and chemoresistance [33,156]. ALDEFLUOR+ melanoma CSCs were linked to chemoresistance; when ALDH1A isoform was silenced, in vitro proliferation was reduced and apoptosis was induced, while in vivo tumorigenesis was inhibited [156]. In xenografted tumors, ALDH1A1 and ALDH1A3 expression was shown to be more than 15 times greater in the ALDEFLUOR+ subpopulation, according to microarray analysis of ALDEFLUOR+/− cells [156]. These findings imply that ALDH1A1 and ALDH1A3 may be the main isoforms that contribute to the ALDH activity in primary melanoma. However, ALDH1A3 mRNA expression was more than 200 times higher than ALDH1A1 in ALDEFLUOR+ subpopulations isolated from melanoma cell lines. More specifically, ALDH activity in 1205Lu and A375 cells was considerably reduced when ALDH1A3 expression was knocked down, pointing to a possible contribution of ALDH1A3 [156]. In another study, ALDH3A1’s expression was determined by IHC in melanoma patients, along with the expression of programmed death-ligand 1 (PD-L1), and cyclooxygenase-2 (COX-2), while after overexpressing it in melanoma cultures, it was correlated with tumor stemness, epithelial to mesenchymal transition (EMT) markers, and PD-L1 expression [157]. In addition, ALDH18A1 seems to play an important role in melanoma growth through the proline biosynthesis pathway, because when it was knockdowned by siRNA, melanoma cell and xenograft tumor growth was inhibited [158].

***Prostate cancer.*** Low levels of ALDH1A1 are expressed in the basal cell layers of normal prostate tissues, where it coexists with the stem cell marker CD44, according to an IHC study of normal tissues and tissue slices from prostate cancer [159]. High ALDH1A1 expression correlates with lower overall survival, high Gleason score, and high pathologic stage in patients with primary prostate cancer. Studies support that ALDEFLUOR+ prostatic cancer cells play a pivotal role in clonogenicity, tumorigenicity, and metastasis [159,160]. It has been shown that ALDH1A1 expression is directly modulated by the Wnt pathway through the β-catenin/TCF-dependent transcription. Inhibition of the Wnt/β-catenin signaling pathway results in a lower viability of prostate cancer cells, which are characterized by high ALDH1A1 expression [161]. Moreover, ALDH1A1 was the main enzyme in ALDEFLUOR+ PC-3 and LNCaP cell lines, which possess stem cell traits [159]. 

Nevertheless, another study supports that ALDH7A1 and not ALDH1A1 was the primary enzyme isoform determining the high ALDH activity found in prostate cancer cells. By analyzing six primary prostate cancer specimens and eight prostate cancer cell lines, it was indicated that ALDH isoforms with higher expression levels were ALDH3A2, ALDH4A1, ALDH7A1, ALDH9A1, and ALDH18A1. All primary cultured cells as well as the PC-3, PC-3M-Pro4lucBIII, and DU145 cell lines displayed high ALDH7A1 expression [160]. It was found that knocking down ALDH7A1 reduced ALDEFLUOR activity by 21% [162]. Furthermore, it has been demonstrated that ALDH4A1 and ALDH9A1 are the primary enzyme isoforms responsible for ALDEFLUOR activity in prostate cancer tissues, whereas ALDH3A2 and ALDH18A1 are the predominant enzyme isoforms in non-tumor tissues and high-grade prostate intraepithelial neoplasia [160].

***Ovarian Cancer.*** ALDH1A1 overexpression is correlated with chemoresistance in ovarian cancer cells [163,164]. ALDH1A1 is predominantly expressed in mucinous and endometrioid epithelial cancer cells, but not in most of the serous and clear cell cancer cells, as shown by IHC, Western blot analysis, and the ALDEFLUOR assay [165]. The expression of ALDH1A isoforms were also investigated in ovarian endometrioma and human endometrial tissue, including those afflicted by endometriosis, by IHC. In the stroma of the endometrium and in the endometriotic ovarian tissue, ALDH1A isozymes were positively stained with varied expression patterns. While ALDH1A2 was only strongly expressed in the epithelium of endometrioma, ALDH1A1 and ALDH1A3 were highly expressed in the stratum basalis of the endometrium, as well as in the epithelium of ovarian endometrioma, regardless of the menstrual cycle. In the glands of stratum basalis, ALDH1A1 is co-localized with N-cadherin, a hallmark of endometrial epithelial progenitor cells [166].

Unlike ALDH3A1 and ALDH3B1, there is evidence that ALDH1A3, ALDH3A2, and ALDH7A1 are overexpressed in ovarian tumors compared to normal ovarian tissues [165]. Conversely, in another study, they showed that ALDH3B1 had an immunoreactivity of 89% in ovarian tumors, such as serous papillary adenocarcinomas, clear cell adenocarcinomas, endometroid adenocarcinomas, and mucinous adenocarcinomas. Most of these tumors were invasive and moderately to poorly differentiated, whereas ALDH3B1 expression was higher in patients under 60 years old [167].

Although the prognostic role of ALDH1 is quite controversial, meta-analyses revealed that individuals with ovarian cancer who express more ALDH1 had lower overall and progression-free survival, poor prognosis, and clinicopathological characteristics [168,169]. On the other hand, ALDH5A1’s transcription and expression have been suggested to be associated with better overall survival in serous ovarian cancer patients expressing mutated TP53, but not in those expressing wild-type TP53, indicating its crucial role in ovarian cancer progression [170].

***Head and neck squamous cell carcinoma.*** ALDEFLUOR+ HNSCC cells (head and neck squamous cell carcinoma) presented stem cell-like characteristics and high CD44 expression, and were capable of forming primary tumors in immunodeficient animals [171,172]. The existing evidence supports that ALDH1A1 and ALDH1A3 are the main isoforms related to this type of carcinoma. ALDH1A1 is expressed in more than 10% of tumor cells in HNSCC tumor samples and in more than 25% of the cells in Fanconi anemia-head and neck squamous cell carcinoma (FA-HNSCC) tumor tissue samples. Similar findings were obtained from IHC analysis of tumors xenografted with FA-HNSCC [173]. On the other hand, Kurth et al. (2015) have showed that ALDH1A3, at least partially, contributes to ALDEFLUOR activity in HNSCC. Comparatively to ALDH1A1, ALDH1A3 showed a high expression profile in FaDu and Cal33 cells. Further analysis of xenografted tumors derived from ALDEFLUOR+ cells also expressed high levels of ALDH1A3 [174].

***Liver cancer.*** While ALDH1A1 is considered a good marker for CSCs in most types of cancer and is correlated with their “stemness”, this does not apply in the case of liver cancer [21]. In hepatocellular carcinoma (HCC) and hepatoblastoma, ALDH1A1 has been described as the predominant isoform—the one dictating ALDEFLUOR activity—and it was found to have differential expression in CD133+/− HCC cell lines [175]. However, the capacity of HCC cell lines to proliferate and form spheres, as well as the levels of epithelial cell adhesion molecule (EpCAM), an HCC CSCs marker, did not significantly change as a result of ALDH1A1 knockdown [176]. Additionally, patients’ tumors with high ALDH1A1 expression were well differentiated [176]. The levels of ALDH1A1 mRNA in tumor and non-tumor tissues were not substantially different, according to an RT-PCR analysis of 47 patient samples of HCC tumorous and their respective adjacent non-tumorous tissues. Moreover, ALDH1A1 was not co-expressed with stem cell markers such as BMI1, EpCAM, CD13, CD24, CD90, or CD13342, according to IHC analysis [177]. Regarding these, ALDH1A1 would alternatively be a differentiation marker with minimal bearing on the preservation of HCC’s stem cell properties [176,177].

ALDH1A3 was shown to be the main isoform that modulates ALDEFLUOR activity in intrahepatic cholangiocarcinoma [178]. According to an analysis of the mRNA expression of 19 ALDH isoforms in the ALDEFLUOR+/− subpopulations, ALDEFLUOR+ HuCCT1 cells expressed higher levels of ALDH1A3 and ALDH1L1, while ALDEFLUOR+ SUN1079 cells expressed higher levels of ALDH1A3, ALDH1B1, ALDH1A1, ALDH6A1, ALDH1A1, ALDH18A1, ALDH3B2, and ALDH3B1 [178]. In these two cell lines, ALDH1A3 was the main isoform in the ALDEFLUOR+ populations. ALDH1A3 knockdown resulted in lower invasion and migration potential and in a significant increase in sensitivity to the chemotherapeutic agent gemcitabine [178].

***Brain Tumors.*** Glioblastoma (GBM) neutrosphere differentiation can be rescued by Notch signaling pathway activation in an RA-dependent mechanism, demonstrating a functional role for hALDHs in GBM carcinogenesis [179]. Moreover, when a highly invasive glioma stem cell (GSC) line, namely, mesenchymal (Mes), was compared to a less invasive glioma GSC line, namely, proneural (PN), the ALDH activity was found to be eight times higher at the first than the latter [180]. 

In particular, ALDH1A1 and ALDH1A3 have been identified as the primarily expressed isoforms in GBM tumors. Evidence supports that ALDH1A1’s expression is higher in high-grade gliomas than in low-grade gliomas [181], whereas ALDH1A3 is expressed in most GBM cases but not in normal tissues and low-grade gliomas [180]. It was suggested that ALDH1A1 could serve as a GSC marker, because it is highly expressed in ALDEFLUOR+ UC25 cells and primary cultured cells with stem cell-like characteristics [181]. However, RT-PCR analysis implied that ALDH1A3’s expression was 150-fold higher in Mes GSCs than in PN GSCs, while all the other ALDH isoforms were expressed at low levels in both groups of GSCs [180]. Additionally, Mes GSCs’ proliferation, sphere formation ability, and tumorigenicity were sufficiently reduced by inhibiting ALDH1A3 [180]. 

However, Park et al. (2018) showed that knocking down *Aldh1l1* in glioblastoma cells reduced their viability, their ATP levels, and the expression of genes associated with stemness, mesenchymal transition, and invasion, indicating ALDH1L1’s role in glioblastoma’s bioenergetics. Following an analysis of Gene Expression Omnibus (GEO) and TCGA databases, they also found that high expression levels of ALDH1L1 were associated with lower overall survival [182].

## 6. The Role of hALDH Proteins in Resistance to Chemotherapies

Several CSCs have demonstrated high expression levels of ALDH isoforms. ALDHs play a crucial role in promoting resistance against chemotherapy through intracellular inactivation, detoxification and binding of chemotherapeutic agents [183]. 

The oxidizing and deactivating effects of several hALDHs on a number of well-known chemotherapeutic agents such as temozolomide, cyclophosphamide, irinotecan, paclitaxel, epirubicin, and doxorubicin have been demonstrated [183]. It has also been demonstrated that ALDH1A1 binds to and diminishes the effectiveness of cytotoxic drugs such as flavopiridol and some cancer medications that target topoisomerase, such as daunorubicin [184]. Recent research has shown that in addition to ALDH1A1, ALDH3A1, ALDH1A2, ALDH7A1, and ALDH2 are also implicated in chemoresistance [18,185]. A growing number of studies also point to ALDH activity for being essential for the control of intracellular scavenging, which protects both healthy and cancer cells from ROS caused by chemotherapy and radiation therapy [186,187,188]. When ALDH is overexpressed, hazardous aldehydes that build up as a result of radiation and ROS-producing chemotherapeutics such as doxorubicin, paclitaxel, sorafenib, and staurosporine are metabolized [189]. Additionally, ALDH-mediated reduction in ROS can reduce immunogenic cell death and anti-tumor immunity by impairing the effectiveness of drugs that generate ROS, such as cyclophosphamide, mitoxantrone, oxaliplatin, bleomycin, and bortezomib [190]. Oxazophosphorine drugs are also directly inactivated by ALDH1A1 and 3A1 [137]. More specifically, in patients with squamous cell carcinoma or adenocarcinoma of the esophagus, ALDH1 expression levels have been demonstrated to predict responsiveness or resistance to preoperative chemoradiation [191]. ALDH1 overexpression was linked to resistance to therapy, an aggressive phenotype in tumor spheres, and increased expression of genes that confer resistance [156]. Additionally, tumor sphere assays revealed that 5-fluorouracil-resistant esophageal cancers had excessive ALDH1 activity and an aggressive phenotype [191]. A similar investigation was performed on CSCs that were resistant to cytarabine and bortezomib in acute myeloid leukemia [192]. Although bortezomib generally prevented the proliferation of cytarabine-resistant cells, it had no effect on ALDH+ cytarabine-resistant cells, in which there was a higher percentage of ALDH+ cells [192].

Furthermore, cisplatin-resistant mesothelioma was linked to ALDH overexpression, due to the fact that cells became more sensitive to cisplatin when DEAB, a multi-ALDH isoform inhibitor, was given before cisplatin therapy [193]. Similarly, increased ALDH1 levels were linked to the development of locally advanced rectal cancer after radiochemotherapy, with increased ALDH1 levels indicating metastasis and disease refractoriness [194]. Additionally, ALDH+ cells with stem cell traits were suggested to be responsible for the Ewing sarcoma family tumors’ resistance to doxorubicin and etoposide [195].

The effectiveness of specific anticancer treatments, such as the epithelial growth factor receptor inhibitors, namely, erlotinib and gefitinib, has also been hampered by ALDH-mediated resistance [196]. Compared to ALDH1A1− cells, ALDH1A1+ lung cancer cells were more resistant to gefitinib [196]. In addition, a greater percentage of ALDH1A1+ cells were present in gefitinib-resistant lung cancer cells [196]. The poly(ADP-ribose) polymerase 1 (PARP1) inhibitor olaparib was not effective against ALDH+ breast cancer cells because of increased PARP1 expression [197].

Targeting ALDH proteins is one of the potential approaches to combat CSC chemo- and radio-resistance [99]: pre-incubation of lung cancer cell lines with all-trans RA decreases the expression level and the enzymatic activity of ALDH1A1 and ALDH3A1 and makes cancer cells far more sensitive and vulnerable to chemotherapeutic treatments [136].

## 7. Cancer Therapeutic Treatments Based on hALDH Inhibition

Targeting the hALDHs has been attempted to improve cancer treatment and combat therapeutic resistance, by identifying specific and non-specific ALDH inhibitors [28]. Identification of isoform-specific inhibitors has been achieved by targeting the catalytic domain or the oligomerization domain because they are different among hALDH isoforms. On the other hand, discovery of selective multi-isoform inhibitors with minimal off-target effect could be achieved by targeting the NAD(P) binding domain. This domain, while it is unique among other oxidoreductases, is similar among the ALDH family [198,199].

### 7.1. Non-Specific hALDH Inhibitors

A high-throughput search for ALDH2 activity modulators led to the discovery of *Aldi 1-4*, four related compounds with comparable inhibitory characteristics and time-dependent kinetics for several ALDHs. The inhibitory activity of the four compounds as it was measured by IC50 against ALDH1A1, ALDH2, and ALDH3A1 fluctuates from 5.4 to 8.6 μM, 1.7 to 12 μM, and 2.2 to 7.9 μM, respectively [200]. The crystal structures of ALDH2/Aldi-3 (PDBid: 3SZ9) and ALDH3A1/Aldi-1 (PDBid: 3SZB) complexes have shown that this kind of inhibitor forms a covalent bond with the protein active site cysteine (Figure 9A,B) [200]. Later, Kim et al. (2017) developed *Aldi-6*, an inhibitor for ALDH1A1, ALDH2, and ALDH3A1 with the IC50 values being 0.6 μM, 0.8 μM, and 1 μM, respectively. In vitro, treatment with Aldi-6 significantly reduced the viability of HNSCC cells, and when combined with cisplatin, Aldi-6 further reduced tumor burden in vivo [201].

*Citral*, a natural occurring aliphatic monoterpene aldehyde [202], exhibited strong inhibitory effect in breast cancer cells against ALDH1A1, ALDH1A3, and ALDH2 [203,204]. Citral can inhibit ALDH1A3-mediated breast tumor development by inhibiting the enzyme’s capacity to form colonies and control gene expression, as well as by controlling the expression of apoptosis and cell-cycle markers [204,205,206]. Additionally, given the advantages of encapsulation in the in vivo administration of drugs, nanoparticle-encapsulated citral was employed to selectively inhibit the increased tumor development of MDA-MB-231 cells overexpressing ALDH1A3 [204].

It has been shown that *Diethylaminobenzaldehyde (DEAB)*, which is widely known as the most frequently used negative control substance in ALDEFLUOR assay, exhibits competitive inhibition of ALDH1A1, ALDH1A3, ALDH1B1, and ALDH5A1 [198,207,208]. DEAB is a reversible substrate for ALDH1A1 and ALDH3A1, an irreversible inactivator of ALDH7A1 (PDBids: 4X0U and 4X0T; Figure 9C) and a time-dependent, reversible inhibitor of ALDH9A1. Its IC50 value for ALDH1A1 is 57 nM, for ALDH2 0.16 μM, for ALDH1A2 and ALDH1B1 1.2 μM, for ALDH1A3 3 μM, and for ALDH5A1 13 μM [198]. Consequently, DEAB’s efficacy as an anticancer drug is highly related to ALDH expression [209]. In particular, DEAB’s function as an ALDH inhibitor in ovarian, breast, and lung cancer has been the subject of substantial research because it seems to inhibit cancer growth and alleviate tumor burden and metastasis [198,209]. Furthermore, DEAB diminishes ALDH+CD44+ breast cancer stem cells’ (BCSC) chemotherapeutic and radiotherapeutic resistance and the number of CD133+ ovarian CSCs [210,211]. In a recent study, DEAB chemical formula was used as a scaffold and forty DEAB analogues were synthesized in order to investigate their ALDH isoform selectivity and cellular potencies in prostate cancer cells [212]. Three of these analogues (named 14, 15, and 16) showed potent inhibitory activity against ALDH1A3 with IC50 values of 0.63 μM, 0.3 μM and 0.23 μM, respectively, and two analogues (18 and 19) showed potent inhibitory activity against ALDH3A1 with IC50 values of 1.61 μM and 1.29 μM, respectively. More importantly, sixteen analogues showed enhanced cytotoxicity with IC50 values ranging from 10 to 200 μM against three distinct prostate cancer cell lines as compared to DEAB which had IC50 values over 200 μM. The most potent analogues seemed to be 14 and 18, because they were more effective than DEAB against patient-derived primary prostate tumor epithelial cells, either as single drugs or in combination therapy with docetaxel [213].

*Disulfiram (DSF)* is a known anti-alcoholism drug which also exhibits an irreversible pan-ALDH inhibitor (IC50 is 0.15 μM and 1.45 μM for ALDH1 and ALDH2, respectively) [214]. The good therapeutic efficacy of DSF has been demonstrated in vivo and in vitro while clinical studies have shown its significance for patients of certain conditions such as alcohol-related disorders and solid malignancies [215,216,217]. Several studies have demonstrated that DSF may combine with Cu to generate a complex (DSF/Cu), which is more easily absorbed by cells and has cytotoxic effects on a range of cancer cells but not on healthy cells. DSF/Cu may prevent tumors formed by sorted ALDH+ CSCs in vivo as well as ALDH+ NSCLC stem cells in vitro [22,217,218]. Combining DSF with chemotherapy can extend overall survival and progression-free survival in patients with NSCLC [219]. Additionally, preclinical studies have shown that DSF effectively reduces the proliferation and tumorigenicity of 4T1 breast cancer cells by targeting myeloid-derived suppressor cells (MDSCs) and ALDH1A1+ CSCs, respectively, when combined with gemcitabine or a PD-L1 antibody [220]. However, in cell cultures and animal models, DSF/Cu can prevent the spread of breast cancer cells by inducing apoptosis or obstructing EMT. Based on these findings, a Phase II clinical trial of the DSF/Cu combination chemotherapy is in progress (NCT04265274) to assess the therapeutic potential for patients with metastatic breast cancer. In recurrent GBM, ALDH1A3 and ALDH1A1 expression levels are increased and tumor cells are more resistant to temozolomide (TMZ) therapy, according to a recent study [221]. The proteins are involved in the detoxification of reactive aldehydes generated from LPO following TMZ therapy. In a Phase II clinical study (NCT03034135), TMZ combined with DSF/Cu overcomes ALDH1A3-mediated TMZ resistance in GBM patients. Only 4% of patients had dose-limiting side effects, indicating that the medication is well tolerated [222]. To treat newly diagnosed GBM, TMZ coupled with DSF/Cu will be utilized as adjunctive and concurrent chemotherapy (NCT01777919); however, the data have not yet been made public.

*4-dimethylamino-4-methyl-pent-2-ynthioic acid-S-methylester (DIMATE)* is one of the most effective ALDH competitive irreversible inhibitors [223]. It has an IC50 of 5 μM for the ALDH1 and ALDH3 subfamilies and an IC50 of 7 μM for the prostate cancer cell line DU145 [224]. DIMATE has also been shown to exhibit little toxicity on healthy cells and to inhibit tumor development in vivo when administered intraperitoneally in melanoma bearing mice [225].

*Dyclonine*, an oral anesthetic, is demonstrated to be a covalent mild inhibitor of ALDH2 and ALDH3A1 (IC50 for ALDH2 is 35 μM and for ALDH3A1 is 76 μM, respectively). Dyclonine and sulfasalazine together effectively inhibit the development of HNSCC or gastric cancer tumors that have high ALDH3A1 expression. Dyclonine monotherapy, however, is ineffective in vivo [200,226].

*2-[4-(5,7-Dibromo-2,3-dioxo-2,3-dihydroindol-1-ylmethyl)benzyl]isothiourea hydrobromide (KS100)* is a novel, potent, multi-isoform ALDH inhibitor (IC50s at 0.21, 1.41, and 0.24 μM for ALDH1A1, ALDH2, and ALDH3A1, respectively). The development of NanoKS100, a nanoliposomal formulation, has significantly lowered its toxicity in mice. NanoKS100 seems to selectively kill melanoma cells sparing the healthy human fibroblasts and suppressing the growth of xenografted melanoma tumors by 65%. Moreover, decreasing ALDH activity led to increased production of ROS, lipid peroxidation, and the buildup of hazardous aldehydes that resulted in apoptosis and autophagy [227].

### 7.2. Specific hALDH Inhibitors

#### 7.2.1. ALDH1A1 Specific Inhibitors 

**Cpd3** compound is an indolinedione-based analogue with significant inhibitory activity for ALDH1A1 (IC50 is 20 nM) and only minor inhibition for ALDH2 and ALDH3A1. It has been shown that **Cpd3** directly interacts with cysteine residues in the active site of ALDH1A1 [228]. However, no in vivo research utilizing analogues based on indolinedione exists. By using **Cpd3** as a scaffold, the indole group was further optimized to theophylline or benzothienopyrimidine and two unique chemical classes of specific ALDH1A1 inhibitors, **CM026** and **CM037**, were defined (IC50 is 0.8 and 4.6 μM, respectively) [199]. The selectivity of these inhibitors is based on a glycine residue exclusively found in the ALDH1A1’s aldehyde-binding pocket (Figure 10A,B and Figure 11) [199]. It has been shown that **CM037** considerably increased the sensitivity of ovarian cancer cells to cytotoxic effects when combined with cisplatin and disrupted the sphere formation and cell survival of ovarian cancer cells [229]. Combining **CM037** with paclitaxel seemed to sensitize SKOV-3-TR, a paclitaxel-resistant ovarian cancer cell line, whereas monotherapy with either agent was ineffective [230]. 

**NCT-501** belongs to a novel class of theophylline-based analogues which are selective inhibitors for ALDH1A1 (IC50 0.04 μM) [231]. **NCT-501** administration can decrease the ability of HNSCC cells to form spheres and migrate, and it is also cytotoxic to cisplatin-resistant HNSCC cancer cells [232]. Due to hepatic processing before entering the systemic circulation, **NCT-501** has a low oral bioavailability, which restricts its use in oral therapy [231]. When paclitaxel therapy was used in tandem with **NCT-501**, it was found to sensitize SKOV 3 TR cells [230].

**NCT-505** and **NCT-506** are two examples of quinoline-based ALDH1A1 selective inhibitors (ALDH1A1’s IC50 is 7 nM for both **NCT-505** and **NCT-506**) [230]. They both exhibit a decent systemic drug exposure response when taken orally and sensitize SKOV 3 TR cells to paclitaxel therapy when used together [230].

Compound **974** is a small ALDH1A1-selective inhibitor, which was discovered during an attempt to understand how ALDH1A1 controls stemness (IC50 of 0.47 μM). The expression of stemness genes, ALDH activity, and the formation of spheroid and colonies were all considerably reduced after treatment of ovarian cancer stem cells (OCSCs) with **974**. A limited dilution experiment performed in vivo revealed that **974** dramatically reduced CSC frequency. Moreover, senescence and the senescence-associated secretory phenotype (SASP) were significantly downregulated in cells treated with **974**, according to transcriptome sequencing of the treated cells. In functional experiments, it was also verified that **974** reduced the stemness and senescence brought on by platinum-based chemotherapy [233].

Compounds **5k** and **5o** were synthesized as derivatives of *Benzo[d]ox-azol-2(3H)-one* and *2-oxazolo[4,5-b]pyridin-2(3H)-one,* respectively, using a 3D-quantitative structure activity relationship (3D-QSAR) model coupled with scaffold hopping [234]. The compounds are ALDH1A1 inhibitors with good selectivity over ALDH2 and ALDH3A1 (**5k** showed IC50 of 0.026 μΜ, >100 μM and 0.321 μM for ALDH1A1, ALDH2 and ALDH3A1, respectively, and **5o** showed IC50 of 0.028 μΜ, 1.925 μM and 0.220 μM for ALDH1A1, ALDH2 and ALDH3A1, respectively) [234].

#### 7.2.2. ALDH1A3 Specific Inhibitors 

**NR6** compound is a novel ALDH1A3-selective inhibitor that was developed from imidazo[1,2-a]pyridines, which are ALDH inhibitors that had earlier been described and had cytotoxic efficacy against glioblastoma and colorectal cancer cells [235,236,237]. Crystal structure revealed that **NR6** binds to an ALDH1A3’s tyrosine residue that is not conserved (Figure 11), driving the isoform-specific selectivity (Figure 10). Moreover, **NR6** stimulates the downregulation of cancer stem cell markers and has anti-metastatic action in tests of wound healing and invasion, targeting GBM and colorectal cells [238].

The lead substance **MCI-INI-3** was discovered as a selective competitive inhibitor for ALDH1A3 by the successful combination of in silico modelling and in-depth knowledge of protein structure (IC50 of 0.46 μM; Figure 10). According to a cellular thermal shift study using mass spectrometry, ALDH1A3 is the main protein that **MCI-INI-3** binds to in mesenchymal glioma stem cell lysates. It appears that **MCI-INI-3** effectively inhibits ALDH1A3 since its inhibitory impact on the production of retinoic acid is equivalent to that of ALDH1A3 knockout [239].

In a recent study, two more compounds, namely, **ABMM-15** and **ABMM-16**, were discovered to be ALDH1A3 specific inhibitors (with IC50 of 0.23 μM for **ABMM-15** and 1.29 μM for **ABMM-16**). The molecules are benzyloxybenzaldehyde derivatives and their binding modes on ALDH1A3 isoform have been computationally confirmed [213]. Additional findings indicated that neither **ABMM-15** nor **ABMM-16** were significantly cytotoxic to any cell line and co-treatment of each of these compounds with doxorubicin (DOX) on breast cancer lines, MCF7 and MDA-MB-231, and prostate cancer line PC-3 resulted in significantly higher cytotoxic effects than DOX treatment alone [240].

#### 7.2.3. ALDH1B1 Specific Inhibitors 

Recently, Feng et al. have found the first ALDH1B1 selective inhibitors, namely, **bicyclic imidazoliums** and **guanidines** that have similar molecular interactions and potencies when they target the ALDH1B1 active site. Both pharmacophores block ALDH1B1 activity in cells; however, the **guanidines** avoid the **imidazoliums**’ off-target mitochondrial toxicity. Their top isoform-selective guanidinyl antagonists of ALDHs (IGUANAs) are proteome-wide target specific and specifically inhibited colon cancer organoids and spheroids from growing. The ALDH1B1-dependent transcriptome, which contains genes that control mitochondrial metabolism and ribosome function, has also been revealed through genetic and pharmacological perturbations [241].

#### 7.2.4. ALDH1Ai Specific Inhibitors 

During the investigation of structural factors influencing inhibitors’ selectivity for ALDH1A isoforms (ALDH1Ai), two new compounds, named **13g** and **13h**, were identified (**13g** has an IC50 of 80 nM, 250 nM, and 120 nM for ALDH1A1, ALDH1A2, and ALDH1A3, respectively, and **13h** has an IC50 of 270 nM, 480 nM, and 130 nM for each of these enzymes, respectively) [242]. In patient-derived ovarian cancer spheroids, **13h** exhibited synergy with cisplatin, and both **13h** and **13g** decreased CD133+ putative stem cells in a dose-dependent manner. The effectiveness of these substances in vivo, however, has not yet been documented [242].

5-Nitrofurans are of great interest in human medicine as pro-drugs, that is, they need bio-activation in order to exhibit their activity. Especially **nifuroxazide** has been shown to be bio-activated by ALDH1 isoforms, but not ALDH2, into its cytotoxic metabolites. The bio-activation occurs by oxidation and inhibition of ALDH1 enzymatic activity. **Nifuroxazide** was also found to kill ALDH1+ melanoma populations [243].

A novel class of **imidazo[1,2-a]pyridine derivatives** have been reported as ALDH1Ai inhibitors [237], which also effectively target GBM and prostate cancer stem cells. In the case of GBM, the 6-(4-fluoro)phenyl derivative called **3b** decreased the cell viability of proneural (PN-157) and mesenchymal (MES-267 and MES-374) cell lines, derived from patients’ glioma spheres, with the IC50 of 25.2 nM, 63.4 nM, and 2.58 pM, respectively [236]. Moreover, **3b** demonstrated both inhibitory efficacy against PC3 colony-forming efficiency as well as antiproliferative activity in the nanomolar range against the P4E6 and PC3 cell lines, derived from localized and bone metastases of prostate cancer, respectively [244].

**673A** compound is another newly discovered ALDH1Ai inhibitor, with an IC50 of 350 nM for each of the three ALDH1A family members and IC50 of 15 μM for ALDH2 or ALDH3. In CD133+ ovarian CSCs, treatment with **673A** induced necroptotic cell death, caused, in part, by a decrease in oxidative phosphorylation and activation of mitochondrial uncoupling proteins. The ability of tumors to initiate was decreased and tumor eradication in vivo was also increased when ALDH1Ai was combined with chemotherapy [210].

#### 7.2.5. ALDH2 Specific Inhibitors 

**CVT-10216** is a very effective reversible inhibitor of mitochondrial ALDH2 (IC50 of 29 nM) [245] and **ALDH423** is another ALDH2 specific inhibitor found through a target-specific rescoring technique (IC50 of 0.62 μM) [246]. **CVT-10216** and **ALDH423** have not yet been investigated in cancer models [28].

#### 7.2.6. ALDH3A1 Specific Inhibitors 

**CB7** and **CB29** are ALDH3A1-specific, reversible inhibitors (Figure 10). Enzymatic and crystallographic studies have shown that **CB7** (IC50 of 0.2 μM) is competitive for ALDH3A1’s substrate binding site and non-competitive for its cofactor binding site. The antiproliferative effects of mafosfamide in conjunction with 10 μM **CB7** were enhanced over monotherapy with mafosfamide in the ALDH3A1-expressing lung adenocarcinoma cells A549 and GBM cells SF767 [247]. **CB29**, similarly to **CB7**, binds to ALDH3A1’s substrate binding site and exhibits an IC50 value of 16 μM. When combined with mafosfamide, **CB29** has an in vitro therapeutic effect that is comparable to **CB7**’s, although no in vivo research data have been published yet [248].

Recently, a 12-residue peptide was identified as a binding partner of ALDH3A1 by using a random phage display library and peptide ELISA. Enzymatic studies showed that the peptide exhibits a significant inhibitory effect against ALDH3A1’s activity which is comparable to **CB29** inhibition. Furthermore, bioinformatic analysis identified an area close to the substrate binding site as the most probable peptide binding site on the protein [249]. 

#### 7.2.7. ALDH4A1 Specific Inhibitors 

The natural substance **acivicin** (**ACV**) is a covalent ALDH4A1 inhibitor that binds to the cysteine residue on the enzyme’s active site (IC50 for ALDH4A1 of 5.4 μM) and suppresses the HepG2 cell line’s growth. With the use of activity-based protein profiling, **ACV** was discovered to be a specific inhibitor of ALDH4A1. Following these findings, **ACV** analogues were created, such as **ACV1**, which has an IC50 for ALDH4A1 of 0.7 μM and is likewise cytotoxic to HepG2 cells [250].

## 8. Conclusions

Human aldehyde dehydrogenases (hALDHs) constitute a superfamily of 19 isoenzymes which are involved in a plethora of physiological and biosynthetic processes for the organism. Most importantly, they have a major impact in the organism’s detoxification as they are responsible for the oxidation of various endogenous and exogenous aldehydes to their corresponding carboxylic acids. Here, we provide a detailed review of the multiple functions of these enzymes as well as elaborate further on their structures by analysing and comparing features of their primary, secondary, tertiary, and quaternary association.

The hALDH superfamily has been associated with the pathogenicity of several health conditions since long ago and raise great interest as they have also been implicated in cancer pathology. This review summarizes and comprehensively presents recent evidence which correlates the action of ALDHs with different cancer types and resistance to chemotherapy. Moreover, we provide a detailed description of the hALDH inhibitors that have been used so far in experimental and clinical settings in cancer therapy and highlight their potential to be used as biomarkers. Future studies focusing on investigating how ALDHs affect cancer cell survival and protect cancer cells against chemotherapeutic agents along with studying the mechanisms of action of the hALDH inhibitors against various cancer types will likely lead to a better understanding of their role in cancer pathology and may lead to new approaches for the development of more efficient cancer therapeutic regimens.

## Figures and Tables

**Figure 1 cancers-15-04419-f001:**
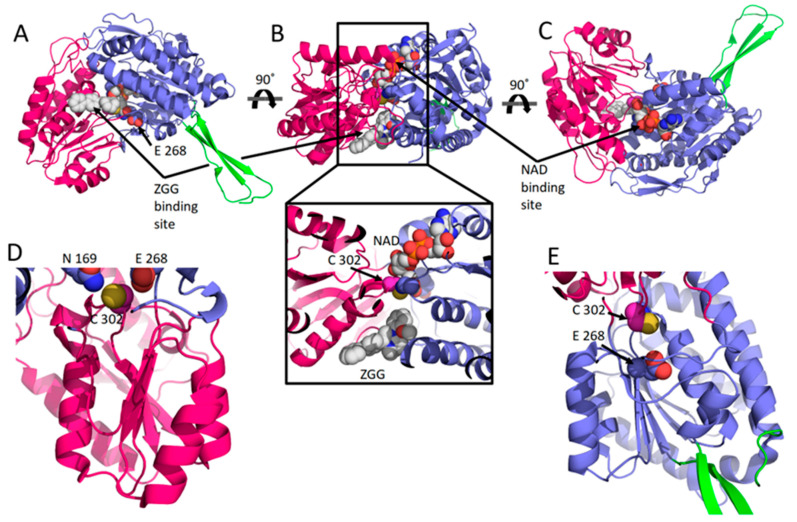
The structure of mitochondrial hALDH1B1 monomer in complex with the NAD (Nicotinamide Adenine Dinucleotide) cofactor and the ZGG (8-(2-methoxyphenyl)-10-(4-phenylphenyl)-1,8-diazabicyclo[5.3.0]deca-1(7),9-diene) inhibitor (PDBid:7MJD). The protein is shown in cartoon representation. Cofactor and inhibitor are shown as space-filling models and indicate the substrate and cofactor binding sites on the protein, respectively. The three catalytic residues, i.e., Cys 302, Glu 268, and Asn 169, are shown with spheres and indicate the protein’s active site. The monomer’s structure consists of a catalytic domain (pink), a cofactor binding domain (blue), and an oligomerization domain (green). (**A**–**C**) Three views of the monomer related with 90-degree rotations around the indicated axis. The zoom-in view of (**B**) focuses on the NAD and ZGG binding sites and thus highlights the spatial proximity of cofactor and substrate binding sites on the protein’s structure. The catalytic (**D**) and cofactor (**E**) binding domains fold as Rossmann motifs. In the interface of catalytic and cofactor binding domains, the active site of the enzyme is located.

**Figure 2 cancers-15-04419-f002:**
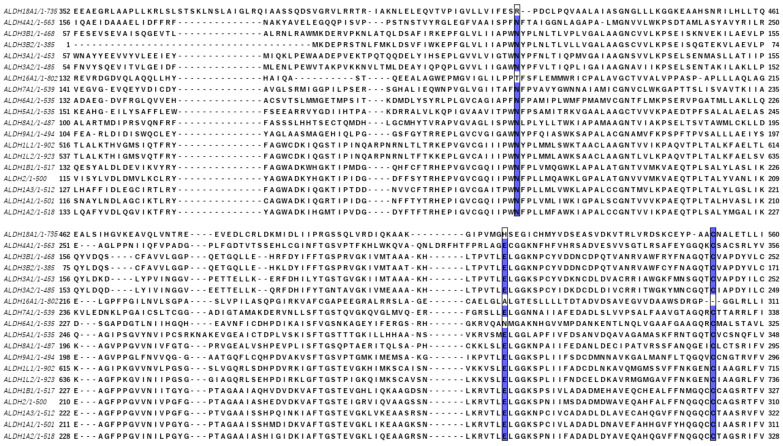
Multiple sequence alignment of the 19 hALDHs. The part of the alignment which includes the catalytic triad (highlighted in blue) is shown. The alignment shows that all hALDHs but ALDH16A1, ALDH6A1, and ALDH18A1 have in common a catalytic triad consisting of a cysteine (catalytic thiol), a glutamic acid (general base), and an asparagine (residue important for stabilizing the reaction’s intermediate). ALDH6A1 follows a slightly different mechanism compared with the other members of the superfamily (see text) and possesses a slightly modified catalytic triad where glutamic acid has been substituted by an asparagine, while the other two catalytic residues (cysteine and asparagine) are conserved. ALDH18A1 is the most distant member of the superfamily (see text and Figure 3) which is also evident from the fact that its active site incorporates only the catalytic cysteine. Last, ALDH16A1 is a pseudoenzyme without enzymatic activity and includes none of the catalytic residues.

**Figure 3 cancers-15-04419-f003:**
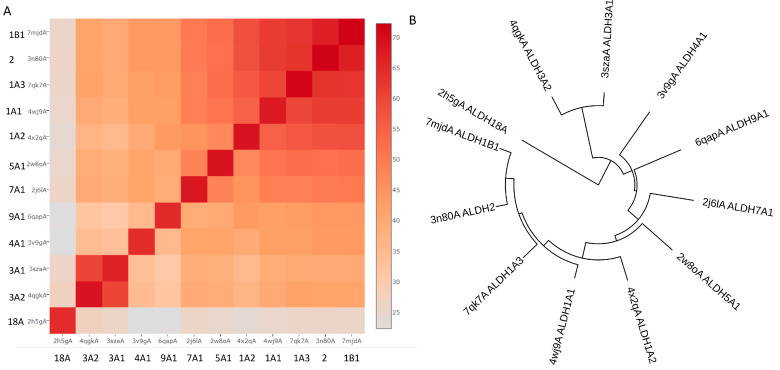
All against all structure analysis of hALDHs with determined 3D structures through DALI server [30]. Protein names and the PDB codes of the structures used as representatives of each protein for the analysis are shown together. (**A**) Heatmap of structural similarity matrix based on Dali Z-scores. (**B**) Structural similarity dendrogram. The dendrogram is derived by average linkage clustering of the structural similarity matrix (Dali Z-scores).The dendrogram figure was prepared with the iTOL tool [31].

**Figure 4 cancers-15-04419-f004:**
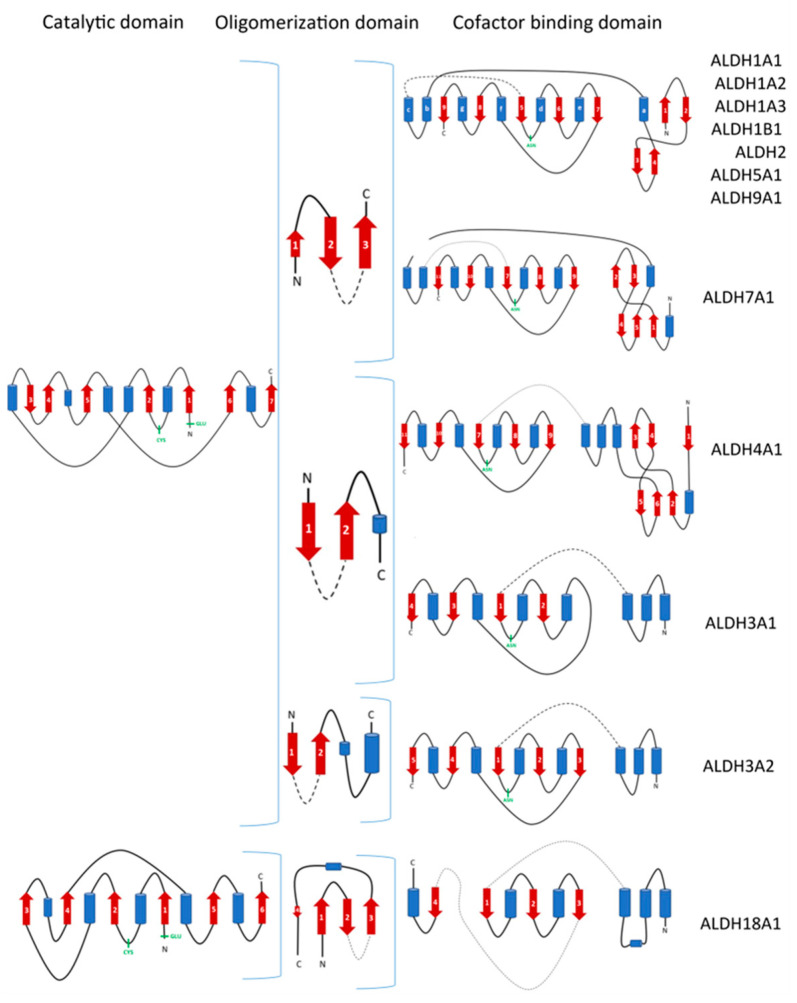
Topology diagrams of the three domains for each of the 12 hALDHs of known structure. All three domains of hALDH18A1 are distinct, while all the other members have in common at least the topology of catalytic domain. hALDH3A1 and 3A2 have in addition a common topology of their cofactor binding domains. ALDH4A1 shares with the hALDH3Ai pair a quite common oligomerization domain. hALDH1/2, 5A1, and 9A1 have common topologies in all three domains, while hALDH4A1 and 7A1 might be the bridge between the 3Ai subgroup and the 1/2, 5A1, 9A1 subgroup.

**Figure 5 cancers-15-04419-f005:**
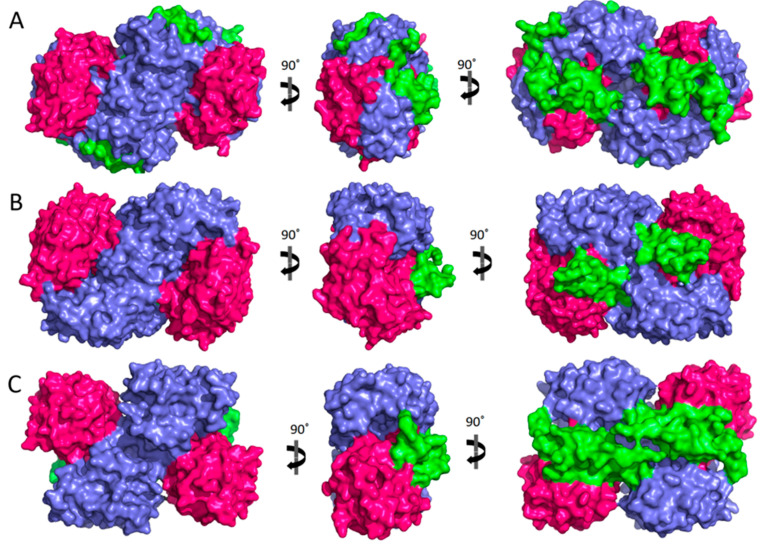
Dimer organization—from three different views, 90 degrees apart—for three representative structures of hALDH superfamily. The structures are shown with surface representation and the colours indicate the catalytic (pink), cofactor binding (blue), and oligomerization (green) domains. (**A**) hALDH3A1 (PDBid: 3SZB). (**B**) hALDH1A1 (PDBid: 4WB9). (**C**) hALDH18A1 (PDBid: 2H5G).

**Figure 6 cancers-15-04419-f006:**
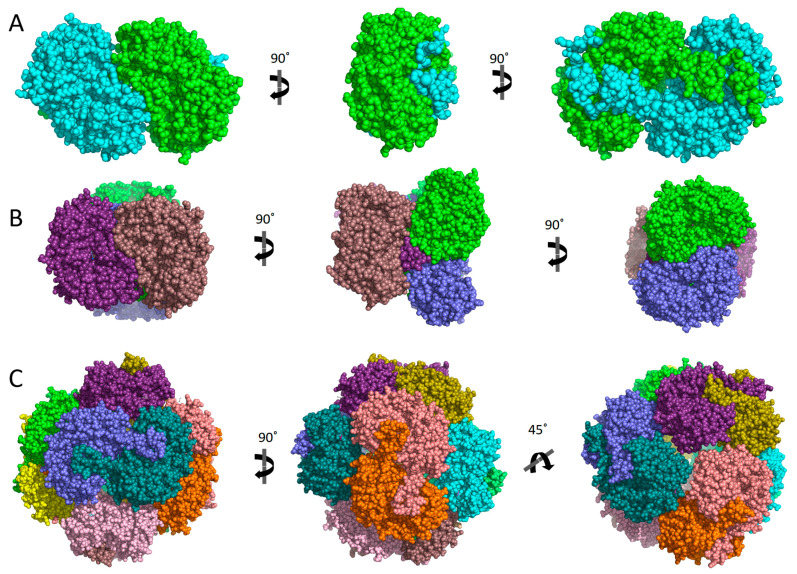
Representative examples of hALDH quaternary assemblies. Each structure is shown from three different views, as indicated. Space-filling models are used, and the different colours represent different monomers. (**A**) hALDH3A1 dimer (PDBid: 3SZB). (**B**) hALDH1A1 tetramer (PDBid: 4WB9). (**C**) hALDH5A1 dodecamer of reduced protein (PDBid: 2W8O).

**Figure 7 cancers-15-04419-f007:**
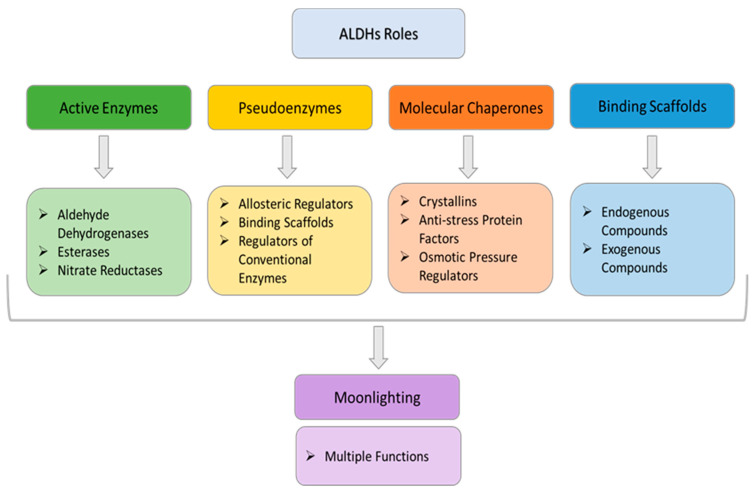
Diagram presenting the multiple roles of ALDHs.

**Figure 8 cancers-15-04419-f008:**
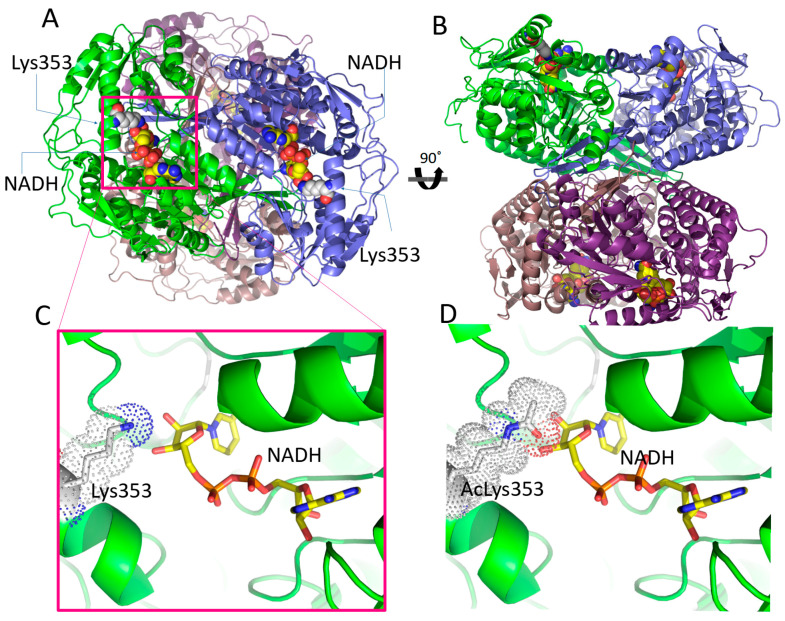
Location of Lysine 353 on the surface of ALDH1A1 tetramer and how its acetylation imposes steric hindrance and affects its catalytic ability. (**A**,**B**) Cartoon representation of hALDH1A1. Each monomer is shown with a different colour. Lysine 353 (gray space-filling model) is found on the rim of NAD(H) (yellow space-filling model) binding pocket. (**C**,**D**) Zoom in on the Lysine 353/NADH binding area. Lysine is shown in a gray sticks model superimposed with van der Waals dots of non-hydrogen atoms. In (**C**), the unmodified Lys makes optimum interactions with the bound NADH. (**D**) A model of acetylated Lys353 shows how the extra acetyl-group restricts the available space and makes unfavourable the NADH binding.

**Figure 9 cancers-15-04419-f009:**
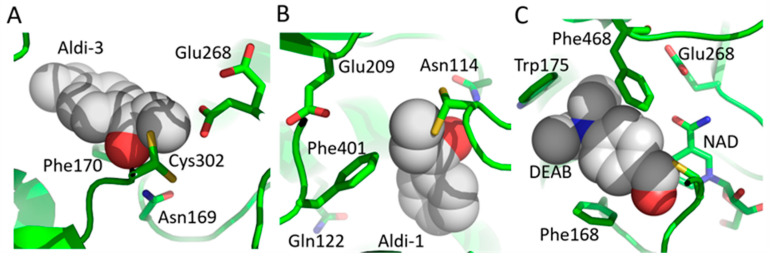
Non-specific inhibitors in complex with hALDHs. (**A**) hALDH2/Aldi-3 complex (PDBid: 3SZ9). The inhibitor is inside the active site and forms a covalent bond with the catalytic cysteine. (**B**) hALDH3A1/Aldi-1 complex (PDBid: 3SZB). (**C**) hALDH7A1/DEAB complex (PDBid: 4X0T).

**Figure 10 cancers-15-04419-f010:**
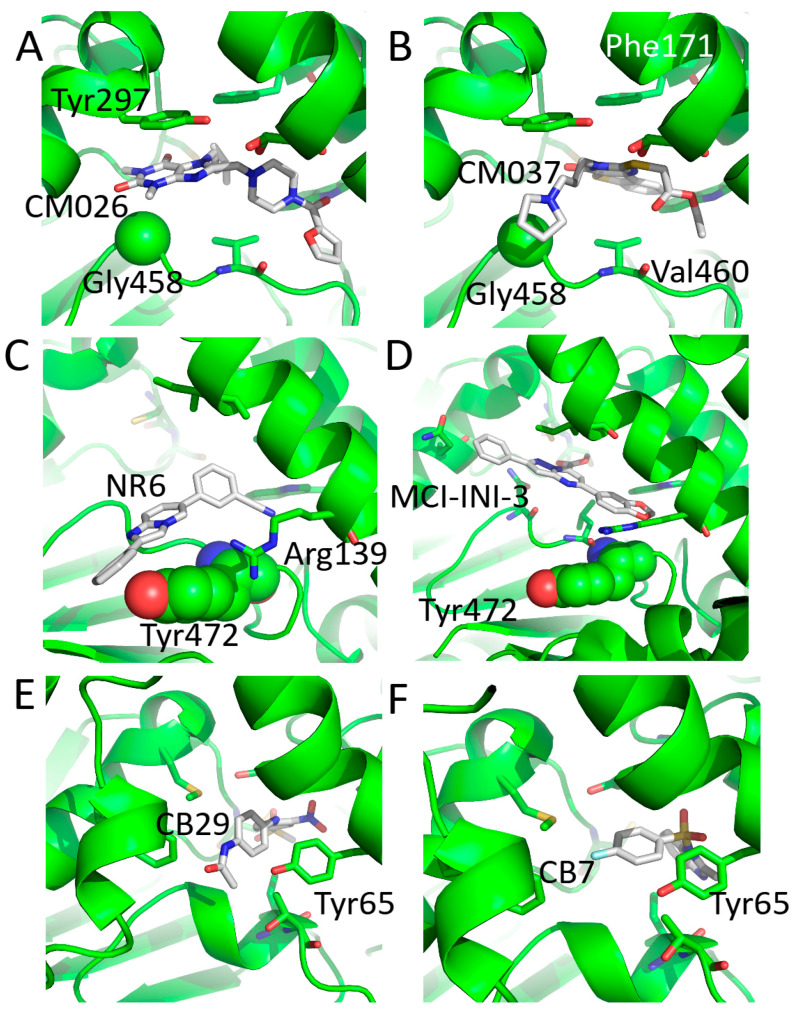
Specific inhibitors in complex with hALDHs. Proteins are shown in cartoon representation and residues significant for the inhibitor-protein interactions are shown with sticks. Especially, non-conserved driving-specificity residues are shown in spacefill. Protein elements are all in green. Inhibitors are represented with gray-carbon stick models. (**A**) hALDH1A1/CM026 complex (PDBid:4WP7). (**B**) hALDH1A1/CM037 complex (PDBid:4X4L). (**C**) hALDH1A3/NR6 complex (PDBid:7A6Q). (**D**) hALDH1A3/MCI-INI-3 complex (PDBid:6TGW). (**E**) hALDH3A1/CB29 complex (PDBid:4H80). (**F**) hALDH3A1/CB7 complex (PDBid:4L2O).

**Figure 11 cancers-15-04419-f011:**
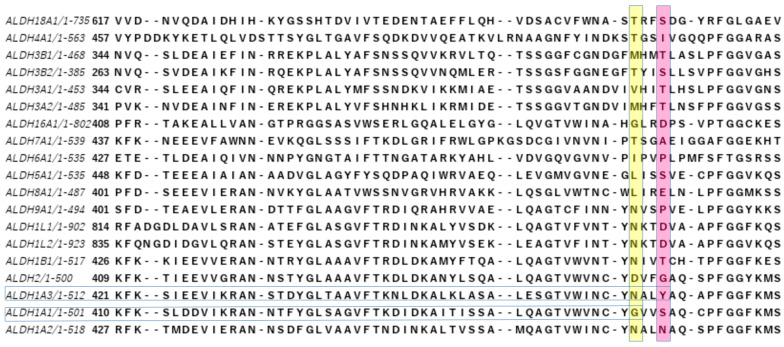
A small window from the sequence alignment of hALDHs, which highlights two residues experimentally characterized as driving-specificity residues for designing inhibitors. Gly458 for ALDH1A1 (last residue of the horizontal yellow highlight) and Tyr472 for ALDH1A3 (last residue of the horizontal pink highlight). For more details, see text and Figure 10.

**Table 1 cancers-15-04419-t001:** Human Aldehyde Dehydrogenases.

**Family**	**Gene/Protein**	**Tissue/Organ** **Distribution**	**Subcellular Localization**	**Quaternary Structure**	**Major Substrate**	**Re****presentative** **PDBs**
1. ALDH1/2	*Aldh1a1*/ Retinal Dehydrogenase 1	Liver, duodenum, stomach, small intestine, erythrocytes, skeletal muscle, lung, breast, other	Cytosol	Tetramer	Retinal	4WJ9, 4WB9, 4WPN
	*Aldh1a2*/ Retinal Dehydrogenase 2	Testis, endometrium, prostate, ovary, other	Cytosol	Tetramer	Retinal	4X2Q, 6B5H
	*Aldh1a3*/ Retinal Dehydrogenase 3	Prostate, bladder, testis, kidney, other	Cytosol	Tetramer	Retinal	5FHZ, 6TGW, 7A6Q
	*Aldh1b1*/ALDHx or ALDH5 or ALDH1B1	Liver, kidney, heart, lung, stomach, other	Mitochondrion	Tetramer	Acetaldehyde	7MJD, 7RAD, 7MJC
	*Aldh2*/ALDH2	Liver, kidney, heart, skeletal muscle, lung, other	Mitochondrion	Tetramer	Acetaldehyde	1O05, 1ZUM (ALDH2*2)
	*Aldh1l1*/FDH or cytosolic 10-formyltetrahydrofolate dehydrogenase or 10-FTHFDH	Liver, kidneys, brain, urinary bladder, skeletal muscle, testis, other	Cytosol	Tetramer	Folate	2BW0 (Hydrolase domain)
	*Aldh1l2*/mtFDH or mitochondrial 10-formyltetrahydrofolate dehydrogenase	Pancreas, brain, stomach, thyroid gland, salivary gland, other	Mitochondrion	Tetramer	Folate	
2. ALDH3	*Aldh3a1*/ALDH3A1	Stomach, skin, cornea, esophagus, other	Cytosol	Dimer	Aromatic aldehydes	3SZA, 4L2O
	*Aldh3a2*/Fatty ALDH	Skin, heart, lung, adrenal glands, kidney, liver, other	Microsome/ ER membrane	Dimer	Fatty aldehydes	4QGK
	*Aldh3b1*/ALDH3B1	Liver, lung, kidney, stomach, breast, other	Cytosol Cell membrane	Unknown	Medium/Long chain aldehydes	
	*Aldh3b2*/ALDH3B2	Skin, esophagus, breast, other	Lipid droplet	Unknown	Medium/Long chain aldehydes	
3. ALDH4	*Aldh4a1*/P5CDH or Delta-1-pyrroline-5-carboxylate dehydrogenase	Liver, kidney, heart, lung, brain, other	Mitochondrion	Dimer	Glutamate γ-semialdehyde	3V9G
4. ALDH5	*Aldh5a1*/SSADH or Succinate-semialdehyde dehydrogenase	Liver, kidney, testis, stomach, heart, brain, other	Mitochondrion	Tetramer (12mer/reduced form)	Succinic semialdehyde	2W8N (oxidized), 2W8O (reduced)
5. ALDH6	*Aldh6a1*/MMSDH or Methylmalonate-semialdehyde dehydrogenase	Liver, kidney, brain, stomach, heart, other	Mitochondrion	Tetramer	Methylmalonate semialdehyde	
6. ALDH7	*Aldh7a1*/AASADH or Alpha-aminoadipic semialdehyde dehydrogenase	Liver, kidney, heart, brain, lung, stomach, other	Cytosol/Nucleus	Tetramer	Betaine aldehyde	2J6L, 4ZUL, 4ZUK
7. ALDH8	*Aldh8a1*/ALDH8A1 or 2-aminomuconic semialdehyde dehydrogenase	Liver, kidney, brain, breast, other	Cytosol	Unknown	Retinal	
8. ALDH9	*Aldh9a1*/ALDH9A1 or TMABA-DH or 4-trimethylaminobutyraldehyde dehydrogenase	Thyroid gland, brain, liver, breast, testis, other	Cytosol	Tetramer	γ-aminobutyraldehyde	6QAK, 6QAO, 6QAP, 6VR6
9. ALDH16	*Aldh16a1*/ALDH16A1	Spleen, duodenum, stomach, kidney, other	Cell membrane	Unknown	Unknown	
10. ALDH18	*Aldh18a1*/ALDH18A1 or P5C Synthetase or Delta-1-pyrroline-5-carboxylate synthase	Small intestine, duodenum, colon, testis, salivary gland, other	Mitochondrion Inner Membrane	Dimer	Glutamatic γ-semi aldehyde	2H5G

* is used to indicate a specific polymorphism of an ALDH isoenzyme.

## Data Availability

No new data were created or analyzed in this study. Data sharing is not applicable to this article.

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
