# Peer review of "Human Aldehyde Dehydrogenases: A Superfamily of Similar Yet Different Proteins Highly Related to Cancer"

_cancers, 2023, doi:10.3390/cancers15174419_

Round 1

Reviewer 1 Report (New Reviewer)

The review is an extensive contribution that recapitulates in an organic manner what reported in literature  in the field of aldhehyde dehydrogenases. In addition, it tries to relate structural features to physiological effect.

Author Response

We would like to thank the reviewer for their time to read our manuscript and their positive comment for our work.

As we understand, the referee does not ask for any changes.

Reviewer 2 Report (New Reviewer)

Xanthis et al wrote a very comprehensive and well documented manuscript about aldehyde dehydrogenases and their  physiopathological implications.

I consider that the manuscript should be accepted for publication in CANCERS.

Author Response

We would like to thank the reviewer for their time to read our manuscript, their positive comment for our work and the excellent marks that they give to us.

As far as we understand, the referee suggests the manuscript's publication as it is.

This manuscript is a resubmission of an earlier submission. The following is a list of the peer review reports and author responses from that submission.

Round 1

Reviewer 1 Report

Vasileios et al presented a review article titled, "Human aldehyde dehydrogenases: a superfamily of similar yet 2 different proteins highly related to cancer". Authors focussed on the implications of ADH enzyme in cancer progression.

Following corrections are to be done during this major revision.

1. Abstract is missing, to be included before introduction with max. of 250 words, covering the methodology of the review, rationale, motivation, and objectives, as well. 

2. References are missing in the first two paragraphs of the introduction section.

3. References cited are mostly old references, authors should use the latest and recent literature reports for the references.

4. authors should explain the main aim and objectives of including the sequence alignment figure 2 and topology diagram figure 4.

5. Conclusion section is missing; the authors must include Conclusion and Future Perspectives. It should be based on your own conclusions from the literature review, and how do you think this direction would go further.